# Potential Therapeutic Targets for Combination Antibody Therapy Against *Staphylococcus aureus* Infections

**DOI:** 10.3390/antibiotics13111046

**Published:** 2024-11-05

**Authors:** Sharon Ke, Hyein Kil, Conner Roggy, Ty Shields, Zachary Quinn, Alyssa P. Quinn, James M. Small, Francina D. Towne, Amanda E. Brooks, Benjamin D. Brooks

**Affiliations:** 1Department of Biomedical Sciences, College of Osteopathic Medicine, Rocky Vista University, Parker, CO 80134, USA; 2Department of Surgery, Virtua Health, Camden, NJ 08103, USA; 3Department of Orthopaedic Surgery, Community Memorial Healthcare, Ventura, CA 93003, USA

**Keywords:** polyclonal antibodies, antibiotic resistance, antibiotics, combination therapies, MRSA, *Staphylococcus aureus*, immunotherapies

## Abstract

Despite the significant advances in antibiotic treatments and therapeutics, *Staphylococcus aureus* (*S. aureus*) remains a formidable pathogen, primarily due to its rapid acquisition of antibiotic resistance. Known for its array of virulence factors, including surface proteins that promote adhesion to host tissues, enzymes that break down host barriers, and toxins that contribute to immune evasion and tissue destruction, *S. aureus* poses a serious health threat. Both the Centers for Disease Control and Prevention (CDC) and the World Health Organization (WHO) classify *S. aureus* as an ESKAPE pathogen, recognizing it as a critical threat to global health. The increasing prevalence of drug-resistant *S. aureus* underscores the need for new therapeutic strategies. This review discusses a promising approach that combines monoclonal antibodies targeting multiple *S. aureus* epitopes, offering synergistic efficacy in treating infections. Such strategies aim to reduce the capacity of the pathogen to develop resistance, presenting a potent adjunct or alternative to conventional antibiotic treatments.

## 1. Introduction

*Staphylococcus aureus* (*S. aureus*) is a Gram-positive, coagulase-positive pathogen commensal to the human body in areas such as the skin, axillary and groin, and mucous membranes of the nose and gut. Many studies show up to 80% of individuals are at least carriers of *S. aureus* [1]. In most cases, this carrier state is asymptomatic, even commensal, but still allows for a reservoir for the pathogen [2]. Commensal organisms that compose the many microbiotas in humans play an essential role in maintaining the health of the many epidermal microenvironments and regulating the immune response. Commensal microorganisms can help prevent colonization by opportunistic and pathogenic bacteria. *S. aureus* frequently shifts to an opportunistic pathogen, capable of causing a wide spectrum of infections, from minor skin conditions to severe, life-threatening diseases such as sepsis [3].

In certain circumstances, such as immunocompromised people or those with disrupted skin barriers, *S. aureus* opportunistically overtakes the skin microbiota, leading to infection and inflammation. When *S. aureus* becomes opportunistic, this can disrupt the balance of the skin microbiome and compromise the skin’s ability to maintain homeostasis and regulate immune responses, leading to skin dermatoses and systemic infections. *S. aureus* is able to achieve this devastation due to numerous mechanisms. The most clinically relevant topic concerning *S. aureus* is infections caused by methicillin-resistant *S. aureus* (MRSA). This pathogenic organism is transmitted in both healthcare and community settings. *S. aureus* is a leading cause for bone and joint infections, endocarditis, skin and soft tissue infections (SSTI), bacteremia, sepsis, and hospital-acquired infections [4,5,6]. For example, *S. aureus* bacteremia has a significant mortality rate of approximately 25% [7]. It also causes many problems due to its historical ability to create serial emergence of epidemic strains [3,8]. Due to this growing amount of genetic diversity and antibiotic resistance, MRSA causes a significant amount of morbidity, mortality, and healthcare costs [4,9]. The average cost of a patient admitted with *S. aureus* bacteremia is greater than USD 12,000.00, with the cost of MRSA being even higher [10]. According to the CDC, MRSA causes ~325,000 infections, resulting in ~11,000 deaths [11]. The estimated annual cost of MRSA infections in the United States is USD 1.7 B, including direct costs (hospital stays, medications, and laboratory tests) and indirect costs (lost productivity and infection prevention and control measures) [11].

One of the significant challenges that *S. aureus* and MRSA present to today’s healthcare system is the pathogen’s ability to acquire and develop antibiotic resistance [5,12]. Broadly, *S. aureus* has intrinsic and extrinsic factors which confer resistance to antibiotic therapy. Examples of intrinsic mechanisms include reduced membrane permeability, efflux systems, and excessive beta-lactamase production. Extrinsic/acquired modalities for antibiotic resistance include its ability to mutate into different forms and acquire resistance genes through plasmid-mediated transduction and transformation, biofilm-mediated resistance, production of persister cells, and additional mechanisms. Because of the myriad strategies and abilities that *S. aureus* possesses to resist antibiotic treatment, it seems necessary to investigate possible targets of immunotherapeutic modalities with which antibiotics could be adjunct to treat effectively.

## 2. Host Immune Response

The immune system defense to *S. aureus* includes both the innate and adaptive system responses. The innate response includes physical and chemical barriers, macrophages, and dendritic cells (DCs). Macrophages and DCs are activated by pattern recognition receptors (PRRs), including toll-like receptors that recognize cell wall biomolecules and organelles glycoproteins, carbohydrates, and flagella. Once activated, macrophages and DCs attack pathogens using phagocytosis, oxidative burst, and complement activation. In addition, they release cytokines and chemokines, increasing the overall immune response and inflammation [13].

The adaptive immune system response to *S. aureus* is more targeted, including both B and T cell responses. T cells play the primary role in orchestrating the adaptive immune response through the release of cytokines, with Th1 and Th17 subsets being particularly important, especially at mucosal surfaces [13]. Th1 cells produce interferon-gamma (IFN-γ), tumor necrosis factor-alpha (TNF-α), and other cytokines that activate macrophages to promote phagocytosis [14]. Th17 CD4+ T cells produce interleukin-17 (IL-17) and other cytokines to recruit neutrophils and promote antimicrobial peptide production on mucous membranes, preventing *S. aureus* colonization. T-cell response to *S. aureus* varies by bacteria strain and the infection’s anatomical location. The lack of clinical success of *S. aureus* therapies, including vaccines, is the reason for developing a CD4+ T cell-mediated immune response [15].

B cells produce antibodies that neutralize specific *S. aureus* antigens [13]. Several effector functions of antibodies can be effective in fighting off *S. aureus* infections, including neutralization (toxins), opsonization (phagocytosis by neutrophils and macrophages), complement (MAC formation), and antibody-dependent cellular cytotoxicity (ADCC) by natural killer (NK) cells. Antibody effector functions all work together, with neutralization and opsonization the most effective [13].

*S. aureus* has evolved numerous immunomodulation and immunoevasion mechanisms, making it difficult for the immune system to control or eliminate the infection. The severity of the infection and the site dictates the response [13]. The complex interplay between the adaptive and innate immune responses against *S. aureus* determines the success of controlling the infection.

## 3. Current Treatment Modalities

MRSA is multidrug-resistant and continues to gain antibiotic resistance at an alarming rate [11]. MRSA has shown drug resistance in several classes, including penicillin, cephalosporins, chloramphenicol, lincomycin, aminoglycosides, tetracyclines, macrolides, quinophthalones, sulfonamides, and rifampicin, which makes treatment in the clinical setting very difficult. Several drugs are found to be more successful in treating MRSA infections in specific settings, and each has its advantages [16]. Here, we will discuss several of the main antibiotics used to treat MRSA currently and their advantages and disadvantages.

Vancomycin has traditionally been the gold standard antibiotic for treating severe MRSA infections. As a bactericidal glycopeptide, it inhibits bacterial cell wall synthesis and demonstrates high efficacy across a wide range of clinical settings. Adverse reactions of vancomycin include ototoxicity and nephrotoxicity, which often requires monitoring [17]. The growing resistance to vancomycin is an increasing concern for the medical community [18]. Resistance to vancomycin is thought to be obtained by binding the drug within the bacterial cell wall. A drug called norvancomycin developed in China has a better concentration-to-effect ratio, which is thought to help with the cost. More studies are needed to assess the clinical effectiveness of norvancomycin definitely [16].

Teicoplanin is another glycopeptide antibiotic similar to vancomycin but is used in patients who cannot tolerate some of the side effects of vancomycin or are allergic. It does have the disadvantage of nephrotoxicity like vancomycin; however, on a milder level. It also can cause allergic reactions, fever, and liver and kidney dysfunction. Teicoplanin is not currently available in the United States [19].

Ceftaroline and ceftobiprole are fifth-generation cephalosporins with demonstrated activity against MRSA [1,20]. They are approved for treating skin and soft tissue infections (SSTI) and community-acquired pneumonia. Both are anti-MRSA cephalosporins that inhibit PBP2a at therapeutically effective concentrations [21]. Ceftobiprole’s R2 group extends into the narrow cleft of PBP2a, allowing access to the active site. In contrast, ceftaroline binding induces an allosteric change in PBP2a, exposing the active site for binding by a second molecule. Adverse effects of both therapeutics include diarrhea, nausea, headaches, and pruritis [5]. Ceftaroline is FDA approved for treatment of MRSA acute bacterial SSTI [22]. The FDA recently approved ceftobiprole medocaril (Zevtera) for adult treatment of *S. aureus* bacteremia, acute bacterial SSTI, and community-acquired bacterial pneumonia. However, ceftobiprole’s broad spectrum of activity of ceftobiprole, particularly against *P. aeruginosa*, can lead to undesirable selection pressure when it is used to treat monomicrobial MRSA infections [20,23].

Infectious Disease Society of America (IDSA) guidelines recommend several alternative therapies for MRSA (Table 1) based on the infection type and mechanism of action [24]. Daptomycin, a lipopeptide antibiotic, is primarily used in severe conditions like bacteremia and infective endocarditis due to its ability to disrupt bacterial cell membranes, leading to rapid cell death [1]. Linezolid, an oxazolidinone, is utilized in treating MRSA pneumonia and works by inhibiting bacterial protein synthesis, making it effective for respiratory infections. Clindamycin, a lincosamide, is often used in pediatric cases and less severe skin and soft tissue infections, functioning by inhibiting protein synthesis at the ribosomal level. These treatments are typically employed in specific clinical contexts where first-line treatments like vancomycin may fail or are contraindicated, with the mechanism of each drug informing its use in standard care [25].

## 4. Description of Targets

*S. aureus*’s life cycle, an evoked immune response, and antibiotic resistance strategies increasingly serve as the basis for identifying potential antibody therapeutic targets. These targets, discussed in the following sections, are critical players in *S. aureus*’s pathogenicity and can be divided into eight functional categories based on their role within the host. Each category highlights important components involved in infection, immune evasion, or resistance to treatment, making them promising candidates for antibody therapy. A comprehensive list of these targets can be found in Figure 1.

### 4.1. Secreted Toxins and Invasins

Many extracellular secreted toxins and invasins act as key virulence factors that manipulate the innate and adaptive immune responses, facilitating the spread of *S. aureus*. These secreted toxins primarily drive inflammation and induce phagocyte death, while invasins aid in the early stages of infection by penetrating host cells and promoting bacterial entry.

Both extracellular toxins and invasins frequently interact with circulating antibodies (e.g., IgM) and secreted antibodies (e.g., IgA and IgM). A promising therapeutic strategy involves developing antibody therapies targeting these virulence factors. This approach may be advantageous because targeting such molecules might not immediately trigger the pathogen’s selective pressure mechanisms, potentially leading to more durable therapeutic efficacy than traditional antibiotics [26,27,28].

The versatility of *S. aureus* as an opportunistic pathogen largely stems from its broad array of virulence factors, many of which are encoded on mobile genetic elements (MGEs) [29]. MGEs, such as plasmids or naked DNA, allow horizontal gene transfer between bacterial species, often providing new evolutionary functions. These elements encode toxins, invasins, adhesins, and immune evasion proteins, discussed later in this manuscript [2,28].

Virulent *S. aureus* molecules that directly harm the host can be categorized into several groups, including pore-forming toxins, exfoliative toxins, membrane-damaging toxins (receptor-mediated and receptor-independent), secreted enzymes, and superantigens [28]. These virulence factors share specific characters: (1) they are universally secreted or expressed extracellularly, (2) they are central players in virulence and pathogenicity, (3) they are responsive to antibody-based interventions, and (4) they are specific to *S. aureus*. These attributes make them highly promising targets for developing therapeutic antibody strategies. Targeting extracellular toxins and invasins by category may significantly reduce virulence, improve clinical outcomes, and help restore normal commensal flora [15].

#### 4.1.1. Hemolysins (Alpha, Beta, Gamma, and Delta)

Hemolysin alpha (Hla or alpha-toxin) is secreted by 95% of *S. aureus* strains. Although not inherently toxic, Hla becomes dangerous through its ability to oligomerize into a heptameric structure on host cell membranes. These protein complexes form a pre-pore structure, subsequently converting into a hydrophilic transmembrane channel, disrupting cellular integrity. Hla targets a broad range of cell types, including epithelial cells, endothelial cells, T cells, monocytes, and macrophages. In erythrocytes, Hla causes skin necrosis and stimulates the release of cytokines and eicosanoids, which can result in systemic shock [30].

Hemolysin Beta (Hlb or sphingomyelinase C) is another toxin produced by *S. aureus* and belongs to the family of pore-forming toxins. Although not fully characterized, Hlb is known to specifically target and damage lipid-rich membranes, particularly those containing sphingomyelin [30]. This toxin exhibits cytotoxic effects on human keratinocytes, polymorphonuclear leukocytes, monocytes, and T lymphocytes, contributing to immune responses, facilitating *S. aureus* survival within host cells. These actions promote phagosomal escape and are linked to biofilm formation, further enhancing *S. aureus*’s ability to persist in chronic infections [28].

Hemolysin Delta toxins are pore-forming toxins that disrupt cell membranes by creating pores, leading to the seepage of the cell and, ultimately, cell death. Leukocidins, in contrast, precisely target and kill white blood cells, which have a key role in the body’s immune response. When both hemolysin Delta and leukocidins are produced by *S. aureus*, they can synergistically damage a broad spectrum of cell types. The leukocidins weaken the immune system by destroying white blood cells, while the hemolysin Delta toxins create pores in various other cell types, causing them to leak and die. This combination of toxins allows *S. aureus* to inflict significant damage on the host, contributing to the severity of infections by compromising immune defense and tissue integrity [26,28].

Isolated gamma-leukocidins play a significant role in dermonecrosis. Gamma-leukocidins are toxins produced by certain strains of *S. aureus* that specifically target and kill white blood cells, particularly neutrophils. Upon production, these toxins trigger the release of inflammatory molecules, such as cytokines and chemokines, which can cause substantial tissue damage and inflammation. The resulting immune response often leads to the formation of abscesses and the destruction of skin and soft tissue, ultimately causing dermonecrosis. Targeting gamma-leukocidins may be a viable therapeutic strategy to mitigate the severe tissue damage associated with *S. aureus* infections [26,28].

#### 4.1.2. Exfoliative Toxin (A and B)

Exfoliative toxins A and B are responsible for skin sloughing in staphylococcal scalded skin syndrome (SSSS) [30]. Although the precise mechanisms of action are not fully characterized, it is speculated that these toxins possess esterase and possibly protease activity [30]. Exfoliative toxins A and B are believed to cause the separation of the outer layer of the skin from the underlying layers, leading to skin sloughing. These toxins are thought to achieve this by cleaving desmoglein-1, a critical protein that holds skin cells together. Additionally, the esterase activity of these toxins allows them to break down certain types of fats, while their potential protease activity may enable them to degrade proteins, further contributing to tissue damage. Early diagnosis and prompt antibiotic treatment are essential for managing SSSS, and in severe cases, supportive care, including fluid and electrolyte replacement, may be required [28,30].

#### 4.1.3. Exotoxins (Pyrogenic Toxin Superantigens—PTSAgs)

Pyrogenic toxin superantigens (PTSAgs) are exotoxins secreted by *S. aureus* that can cause hypotension and cytokine storms [26]. The three basic biological properties exhibited by the exotoxins—pyrogenicity, superantigenicity, and enhancement of endotoxin lethality—are seen as the cause or are moderately involved in many acute or chronic disease pathogeneses [26].

Superantigenicity, the best-characterized property, is the ability to stimulate nonspecific T lymphocyte proliferation by binding to the T-cell receptor (TCR) beta chain variable portion (V beta), which can lead to cytokine storms [26]. PTSAgs-induced hyper-sensitivity towards endotoxins is speculated to be due to reduced hepatic clearance of circulating endotoxins. The increased endotoxin levels release lethal monokines (e.g., TNF-alpha) from macrophages [26]. In addition to promoting the release of vasoactive mediators, PTSAgs can cause hypotension by binding to receptors on endothelial cells and cause capillary leakage [26]. Currently, the PTSAgs family includes TSST-1 and the majority of staphylococcal enterotoxins [26].

#### 4.1.4. Enterotoxins (A-E, G-X)

Staphylococcal enterotoxins (SEs) are potent emetic agents responsible for staphylococcal food poisoning (SFP), among the most common causes of foodborne illness in the United States [26]. When introduced into the body through non-gastrointestinal routes, SEs can also lead to Toxic Shock Syndrome (TSS) unrelated to menstruation [30]. These toxins are mainly produced as small, non-glycosylated polypeptide molecules during the post-exponential growth phase. The genes encoding SEs are transported via MGEs such as plasmids, bacteriophages, or pathogenicity islands. Their expression levels are regulated by at least three global regulatory systems: the accessory gene regulator (agr), the staphylococcal accessory gene regulator (sar), and a catabolite repression system. Experimentally, PTSAgs, including SEs are found to be adequately stable to chemical inactivation, proteolysis, and denaturation via boiling. Producing antibodies against a specific SE does not guarantee cross-protection against other SE variants and does not provide immunity to SFP. Therefore, targeting multiple SEs with antibody therapy is crucial for building immunity against SFP and TSS.

#### 4.1.5. Toxic Shock Syndrome (Toxin-1 (TSST-1))

Toxic Shock Syndrome Toxin-1 (TSST-1) is encoded by the tstH gene, which resides within the staphylococcal pathogenicity island [26]. The translated precursor protein consists of 234 amino acids, with a 40-amino-acid signal sequence cleaved during maturation. The mature protein exhibits a high proportion of hydrophobic amino acids, making it highly soluble in water despite lacking cysteine residues and resistant to heat and proteolysis [26]. Structurally, TSST-1 contains two domains: a long central alpha-helix surrounded by a five-strand beta-sheet and barrel motif. Studies investigating its superantigenic and lethal properties revealed that these functions are separable [26].

Toxoid vaccines may be helpful for individuals who fail to develop immunity to TSS through natural exposure. TSST-1 can cross mucosal surfaces and activate large populations of T lymphocytes simultaneously by cross-linking with TCR V beta domains on major histocompatibility complex (MHC) class II molecules [31]. TSST-1 has also been implicated in various clinical conditions, including Sudden Infant Death Syndrome (SIDS), Kawasaki syndrome, TSS from intravaginal sources, and bacterial cell wall-induced arthritis [32].

#### 4.1.6. Microbial Surface Component Recognizing Adhesive Matrix Molecules (MSCRAMMs)

Microbial Surface Component Recognizing Adhesive Matrix Molecules (MSCRAMMs) are a class of molecules externally expressed for mediating *S. aureus* binding to the host’s extracellular matrix [30]. MSCRAMMs are covalently linked to the cell wall peptidoglycan [33]. MSCRAMMs are organized in the following sequence: N-terminal signal peptide, an exposed ligand-binding domain, directly repeated sequences, characteristic hydrophobic cell wall- and membrane-spanning domain, a C-terminal LPXTG motif (cell-wall anchorage), and a positively-charged tail [33]. Bound host proteins include fibronectin, fibrinogen/fibrin, elastin, vitronectin, collagen, laminin, decorin, and heparin sulfate-containing proteoglycans [34]. MSCRAMMS were explored in the 1980s fragments to immunize mice, resulting in a decreased mortality rate of 87% to 13% [35]. Another study found that fibronectin-binding protein immunization resulted in a 2-log (99%) decrease in bacterial density in experimental endocarditis [36]. A potentially potent protective measure would be a vaccine containing multiple MSCRAMM components.

#### 4.1.7. Clumping Factor A (ClfA)

Clumping factors are specialized surface proteins that bind fibrinogen, leading to the characteristic aggregation of *Staphylococcus* cells when mixed with plasma [30]. Clumping factors A (ClfA) and B (ClfB) contain a repeating sequence of serine-aspartate dipeptides between their ligand-binding domain and the region spanning the bacterial cell wall [33]. Divalent cations, such as Ca^2^⁺ and Mn^2+^, modulate the interaction of ClfA and ClfB with fibrinogen [37,38]. Although the precise role of coagulase and clumping factors in pathogenesis remains unclear, they contribute to immune evasion by inducing localized clot formation, which may facilitate bacterial adherence to injured tissue, endothelial cells, and foreign materials [30]. Ongoing research is exploring the potential of clumping factors as targets for vaccine development [30].

ClfA is a 92 kDa MSCRAMM located on the surface of *S. aureus*, where it binds to both fibrinogen and fibrin [30]. This interaction promotes the clumping of bacterial cells and enhances their adherence to blood clots, plasma-coated biomaterials, and damaged heart valves [30]. The fibrinogen-binding region of ClfA has been identified between residues 332 and 550, with Glu526 and Val527 playing critical roles in binding the C-terminus of the fibrinogen gamma chain [39,40]. The binding activity of ClfA is regulated by Ca^2^⁺ and Mg^2^⁺, which interact with an EF-hand motif, inhibiting its binding to fibrinogen [38]. Mutant strains lacking functional ClfA fail to form clumps in the presence of soluble or adhered fibrinogen [33]. Substitution mutations at residues involved in binding the C-terminal four amino acids of the fibrinogen gamma chain—Tyr^256^, Pro^336^, Tyr^338^, and Lys^389^—greatly diminish ClfA’s affinity for fibrinogen [41]. In animal studies, passive transfer of anti-ClfA antibodies protected mice from septic arthritis and death, while a DNA vaccine encoding ClfA, combined with genetic adjuvants, induced IgG2 anti-ClfA antibodies and reduced *S. aureus* adherence to mammary gland epithelial cells [42,43]. Further research in at-risk patient populations is needed to validate its protective effects, but ClfA remains a promising candidate for partial or complete protection against invasive *S. aureus* infections [30].

#### 4.1.8. Clumping Factor B (ClfB)

Clumping Factor B (ClfB) is a 124 kDa surface protein that binds to fibrinogen, leading to platelet aggregation and interacting with type 1 cytokeratin, specifically K10, found on desquamated human epithelial cells [33,40]. ClfB is associated with the gene clfB and binds to the alpha-chains of fibrinogen [44]. Affinity towards cytokeratin K10 and human desquamated nasal epithelial cells indicate that ClfB may play an essential role in *S. aureus* nasal colonization [40]. In mice studies, *S. aureus* with deficient ClfB had decreased nasal colonization, making it an attractive vaccine component [45]. Further studies are needed to differentiate whether ClfB only decreases nasal colonization or decreases asymptomatic colonization on other parts of the body [30].

#### 4.1.9. Serine-Aspartate Repeat-Containing (Sdr) Protein Family

ClfA and ClfB are also classified as members of the Sdr protein family due to their repeating serine-aspartate (SD) regions. Other notable proteins in this family include SdrC, SdrD, SdrE, and plasmin-sensitive (PIs), which share structural similarities with ClfA and ClfB [33]. SdrC binds to beta-neurexin, and both SdrC and SdrD contribute to adherence to human desquamated nasal epithelial cells [46,47]. SdrE, on the other hand, is known to induce platelet aggregation [48].

#### 4.1.10. Plasmin-Sensitive Protein

Plasmin-sensitive (PIs), a potential virulence factor, binds to cellular lipids and glycolipids and is a covalently linked surface protein involved in intercellular adhesion on nasal epithelial cells and bacterial cell aggregation [49,50,51]. Pls also plays a protective role by preventing the binding of *S. aureus* to IgG, fibronectin, and host cell internalization by acting as a steric hindrance [52].

#### 4.1.11. Bone Sialoprotein-Binding Protein (Bbp)

Bone sialoprotein-binding protein (Bbp) is a 97 kDa protein that binds explicitly to bone sialoprotein, a glycoprotein found in the extracellular matrix of bone and dentine [53]. Bbp was detected in *S. aureus* strains linked to bone and joint infections. Its immunogenic nature and active expression during infection point to a potential role in the onset of osteomyelitis [51].

#### 4.1.12. Degradation Enzymes

Hyaluronidase is a virulence factor that degrades hyaluronic acid, a significant component of the host’s extracellular matrix. By breaking down hyaluronic acid, hyaluronidase allows bacteria to invade deeper into tissues, facilitating the spread of infection and contributing to conditions such as cellulitis and abscess formation. Often referred to as a “spreading factor”, hyaluronidase enhances bacterial dissemination within host tissues, aiding in immune evasion and biofilm formation. By enabling the breakdown of structural barriers, hyaluronidase indirectly contributes to the severity of infections such as necrotizing fasciitis and deep tissue abscesses [54].

Fibrinolysin, also known as staphylokinase, is an enzyme that facilitates the breakdown of fibrin clots. This enzyme works by activating plasminogen to break down the fibrin mesh that forms clots. Once the clot is dissolved, *S. aureus* spreads more easily to other tissues. This enzymatic activity is crucial for disseminating the bacteria. *S. aureus* also produces coagulase to form protective fibrin clots around itself. Fibrinolysin works in tandem to break these down when the bacteria need to escape from the clot and spread to other areas. This action is especially important in conditions like sepsis, where bacterial spread can lead to systemic infection, contributing to the severity of infections [54].

Lipase increases the survival of *S. aureus* on lipid-rich surfaces like human skin and sebaceous glands and has the ability to invade tissues by breaking down lipids in the host’s cell membranes [55]. By breaking down the lipid layers of host cells, lipase enables the bacteria to invade deeper tissues and evade the immune system. Lipid hydrolysis may play a role in stabilizing the biofilm matrix and supporting bacterial colonies in nutrient-limited environments. This enzyme is particularly interesting in studying skin and soft tissue infections. Lipase is a potential target for therapeutic strategies to prevent or limit *S. aureus* colonization and invasion [55].

### 4.2. Secretion System Proteins

*S. aureus* employs several specialized secretion systems that play critical roles in its pathogenesis by facilitating the export of various virulence factors, toxins, and enzymes, and they can serve as targets for therapeutics. These systems include the Type VII Secretion System (T7SS), Type I Secretion System (T1SS), the Sec Pathway, and the Tat System [56,57]. Secretion systems allow *S. aureus* to invade host tissues more efficiently, evade immune responses, and establish infections. By utilizing these secretion systems, *S. aureus* can persist in various environments, contribute to biofilm formation, and ultimately enhance its ability to cause chronic infections [56,57].

These secretion systems may appear promising targets for antibody therapies due to their crucial role in pathogenesis and membrane localization. However, a significant challenge lies in the small transmembrane protein domains, often insufficiently exposed for effective antibody binding. Additionally, the thick peptidoglycan layer in *S. aureus* acts as a physical barrier, preventing antibodies from accessing these transmembrane components. As a result, while the secretion systems are integral to bacterial survival, targeting them with antibodies is complicated by limited extracellular accessibility and structural shielding provided by the cell wall.

#### 4.2.1. Sec and Accessory Sec Pathways

The Sec pathway is the primary protein translocation system in *S. aureus* and other bacteria, responsible for transporting unfolded proteins from the cytoplasm across the inner membrane [58]. The Sec pathway machinery is composed of a multi-protein complex, including SecYEG, which forms the core translocon through the membrane, and SecA, an ATPase that powers the transport by providing the necessary energy through ATP hydrolysis. The Sec pathway recognizes proteins marked with a signal peptide at their N-terminus, guiding them through the translocon channel. Once transported across the membrane, the signal peptide is cleaved, and the protein is secreted outside the cell. This system plays a vital role in exporting various virulence factors, including adhesins, toxins, and enzymes, allowing the bacterium to colonize and infect host tissues. The Sec pathway is minimally accessible to antibodies, making it a high-risk, high-reward target [58].

The Sec accessory pathway in *Staphylococcus aureus* is a specialized variation of the Sec system that assists in translocating specific lipoproteins and other virulence factors across the bacterial membrane [59]. Structurally, it involves additional components such as SecDF, which acts as a chaperone and energy provider to aid in protein folding and secretion, working in conjunction with the core Sec machinery (SecYEG) [59]. Just like the Sec pathway, the Sec accessory pathway is a high-risk, high-reward target due to its minimal accessibility by antibodies.

#### 4.2.2. Type I Secretion System (T1SS)

The Type I Secretion System (T1SS) in *S. aureus* is a one-step transport mechanism that directly moves proteins from the cytoplasm to the extracellular space without requiring a periplasmic intermediate step [60]. This system consists of three key components: an ABC transporter located in the inner membrane, a membrane fusion protein (MFP), and an outer membrane protein (OMP). T1SS primarily exports large, folded proteins, like hemolysins and proteases, contributing to bacterial virulence by lysing host cells and degrading tissue barriers. The ABC transporter, located in the inner membrane, has minimal extracellular exposure, with most of its domains buried within the lipid bilayer, limiting its accessibility to antibodies. Similarly, the membrane fusion protein (MFP), which spans the periplasmic space and links the inner and outer membrane components, also has only small, minimally exposed regions that antibodies could potentially target [60].

#### 4.2.3. Type VII Secretion System (T7SS)

Although the Type VII Secretion System (T7SS) remains incompletely characterized, research has established its essential role in *S. aureus* virulence and competitive advantage [61]. The Type VII Secretion System (T7SS) in *S. aureus* is a complex, multi-protein structure responsible for the secretion of various virulence factors, contributing to bacterial survival, competition, and immune evasion. The core components of the T7SS include EsaA, EssA, EssB, EssC, and EssE, which form the transmembrane complex. These proteins facilitate the passage of effector proteins, such as EsaD and EsxA, from the bacterial cytoplasm into the extracellular space [61].

The T7SS in *S. aureus* relies on several essential membrane proteins to transport virulence factors [56]. EsaA is an integral membrane protein that anchors and stabilizes the secretion machinery, playing a pivotal role in forming the translocation channel. EssA and EssB also contribute to this channel’s structure, ensuring the proper translocation of effector proteins like EsaD across the inner membrane. EssC, an ATPase, provides the energy required for the secretion process by hydrolyzing ATP, driving the movement of these effector proteins through the T7SS. Meanwhile, EssE functions as a scaffold, interacting with other components to maintain the system’s structural integrity. Together, these proteins create a coordinated complex that allows *S. aureus* to secrete virulence factors into the extracellular space, enhancing its ability to compete with other bacteria and evade host immune defenses [56]. These proteins are minimally accessible to antibodies primarily due to the thick peptidoglycan layer. Each of these proteins has small protein domains that could serve as a high-value target; however, the small target and thick layer of carbohydrates would make the endeavor high risk (Figure 2).

#### 4.2.4. Tat Translation System

The Tat system is responsible for transporting fully folded proteins, including those bound to tightly associated cofactors [62]. Target proteins are recognized by the system through a twin-arginine (RR) motif in their signal peptides. Once identified, they are guided through a translocon complex composed of TatA, TatB, and TatC proteins. Enzymes and other proteins involved in critical processes such as metabolism, virulence, and stress responses are exported by this pathway. While not as prominent as other secretion systems, the Tat system’s role in pathogenesis supports the bacterium’s survival ability in challenging environments, such as during nutrient limitation or oxidative stress, and can contribute to biofilm formation and host tissue invasion [62]. Again, the Tat system is a high-risk/high-reward target as specific loops or regions may extend into the periplasmic space, but the Tat proteins remain largely inaccessible to antibodies.

### 4.3. Quorum Sensing/Metabolites

Quorum sensing in *S. aureus* is a sophisticated cell-to-cell communication system that enables the bacteria to coordinate behavior in response to population density [63]. This regulatory mechanism governs the expression of virulence factors, biofilm formation, motility, and antibiotic resistance, making it essential for the bacterium’s ability to adapt to varying environments and establish infections. These systems allow *S. aureus* to synchronize its pathogenic behavior, ensuring survival and persistence in hostile environments, particularly during chronic infections. The primary quorum sensing system in *S. aureus* is the Accessory Gene Regulator (Agr) system, which uses autoinducing peptides (AIPs) to mediate cell communication [64,65,66]. Additional systems, such as the LuxS pathway and the Type I Arginine Catabolic Mobile Element (ACME), contribute to quorum sensing and the bacteria’s metabolic adaptability. Other quorum-sensing systems exist but are minor or very poorly immunogenic [64,65]. Most quorum-sensing molecules are small and thus poorly immunogenic.

#### 4.3.1. Autoinducing Peptides (AIPs)

Autoinducing Peptides (AIPs) are small, secreted signaling molecules from the Agr system involved in quorum sensing [64,65]. AIPs are central in quorum sensing, regulating bacterial communication, and coordinating group behaviors like virulence and biofilm formation. Specifically, AIPs function as extracellular signaling molecules that enable bacterial cells to sense and respond to population density [64,65]. AIPS are generally considered weakly immunogenic due to their small size and peptide nature. Synthetic AIPs or modified versions of these peptides might be more immunogenic if designed for therapeutic or vaccine purposes. The other consideration is that these molecules are commonly found in biofilms, which may hinder antibody access.

#### 4.3.2. Phenol-Soluble Modulins (PSMs)

Phenol-Soluble Modulins (PSMs) are a group of small, amphipathic peptides that are produced and tightly regulated by the Agr system [67]. These peptides play a crucial role in the bacterium’s virulence by contributing to biofilm formation, dispersal, and immune evasion [68,69,70]. PSMs promote biofilm structuring by forming channels that allow nutrient flow and waste removal, and they also facilitate the dispersal of biofilm cells to colonize new areas [71]. Additionally, PSMs have cytolytic properties, enabling them to destroy host immune cells, such as neutrophils, thus evading the immune system. Due to their surfactant-like properties, PSMs also contribute to spreading motility and the ability of *S. aureus* to move across surfaces. PSMs are generally considered weakly immunogenic because of their small size and amphipathic structure [69].

However, PSMs can still interact with the immune system under certain conditions. For instance, PSMs have been shown to trigger pro-inflammatory responses by activating immune receptors like Toll-like receptor 2 (TLR2), though these responses are generally not as strong as those elicited by larger, more structurally complex proteins [72]. This limited immunogenicity, combined with their cytotoxic properties, allows PSMs to play a dual role in helping *S. aureus* evade the immune system while still exerting its pathogenic effects.

#### 4.3.3. Autoinducer-2 (AI-2)

The LuxS system produces Autoinducer-2 (AI-2), a signaling molecule involved in interspecies communication. Unlike the Agr system, which is specific to *S. aureus*, the LuxS/AI-2 system is present across a wide range of bacterial species, allowing it to mediate interactions between different bacterial populations. AI-2 plays a key role in biofilm formation by promoting the structural integrity of biofilms and influencing bacterial community behavior. Additionally, AI-2 contributes to the regulation of metabolism, helping bacteria adapt to environmental changes such as nutrient availability. Though its role in *S. aureus* is less prominent than the Agr system, AI-2 is critical for interspecies signaling, impacting both virulence and microbial ecosystem dynamics [73]. The immunogenicity of AI-2 molecules is similar to that of AIPs; they are small and do not provoke a strong immune response, but they may be therapeutically engineered for increased immunogenicity.

#### 4.3.4. ACME (Type I Arginine Catabolic Mobile Element)

The Type I Arginine Catabolic Mobile Element (ACME) promotes arginine catabolism, aiding bacterial survival in hostile environments, particularly during immune responses [74]. ACME helps *S. aureus* neutralize acidic conditions and resist nitric oxide (NO), an immune defense molecule, by reducing its levels. This system enhances quorum sensing by linking metabolic adaptation to regulating virulence factors, increasing the bacterium’s ability to thrive under immune stress [64,65]. ACME-related components, such as enzymes in the arginine deiminase pathway, are generally considered weakly immunogenic. Since they are primarily involved in metabolic processes rather than being surface-exposed antigens, these proteins are less likely to elicit a robust immune response [64,65]. However, in particular therapeutic or vaccine contexts, immunogenicity could be enhanced by conjugating ACME components with more immunogenic molecules.

### 4.4. Antibiotic Resistance Determinants

Antibiotic resistance in *S. aureus* is driven by various molecular mechanisms that allow the bacterium to evade the effects of widely used antibiotics [75]. These mechanisms involve intracellular enzymes and membrane-bound proteins, many of which are not easily accessible to the immune system. Key resistance determinants include the mecA gene, which encodes PBP2a, a protein that confers resistance to β-lactam antibiotics, and the erm genes, which modify the bacterial ribosome to resist macrolides. Additionally, tetK and vanA contribute to resistance against tetracyclines and vancomycin through efflux pumps and cell wall modifications. While most of these resistance factors are either intracellular or membrane-bound and, therefore, inaccessible to antibodies, exceptions such as beta-lactamase, secreted extracellularly, present potential targets for therapeutic interventions [76]. Understanding the accessibility of these proteins to antibodies is critical for developing new treatment strategies to combat resistant *S. aureus* infections.

#### 4.4.1. Beta-Lactamase

Beta-lactamase breaks down β-lactam antibiotics such as penicillins and cephalosporins, rendering them ineffective. It achieves this by hydrolyzing the β-lactam ring structure of these antibiotics, preventing them from inhibiting bacterial cell wall synthesis. Because beta-lactamase is secreted extracellularly, it is accessible to antibodies, making it a potential target for antibody-based therapies [77]. Targeting beta-lactamase directly could help restore the effectiveness of β-lactam antibiotics in resistant strains [78].

#### 4.4.2. Bifunctional Transglycosylase-Transpeptidase PBP2

PBP2a, encoded by the mecA gene, is a membrane-associated protein responsible for conferring resistance to β-lactam antibiotics [79]. Due to its low affinity for these antibiotics, this protein plays a primary role in cell wall synthesis, even in the presence of β-lactams. PBP2a is primarily located within the bacterial cell membrane. While its active site is intracellular, some external domains of the protein are exposed during bacterial cell division, making them potentially partially accessible to antibodies [79].

#### 4.4.3. Tetracycline Efflux Pump (TetK)

TetK, a tetracycline efflux pump, actively exports tetracycline out of the bacterial cell, preventing the antibiotic from reaching its intracellular target and rendering it ineffective. As TetK is a membrane protein, most of its structure is embedded within the bacterial membrane, with only certain surface-exposed domains potentially accessible to antibodies [80]. However, most of the protein remains hidden within the lipid bilayer, making it minimally accessible for direct antibody targeting.

### 4.5. Motility Factors

Two types of motility have been identified in *S. aureus*: spreading and comet formation. Motility is often an adaptation required for both survival and dissemination. The mechanism of spreading motility for *S. aureus* is considered a passive motility, while comet motility resembles an active motility that requires energy. Active motility (e.g., comet) is defined by an energy-dependent mechanism that allows the bacteria to control where it goes [80]. Passive motility, such as spreading, occurs when bacteria modify their environment to facilitate movement. The two forms of motility observed on agar, spreading and comet formation, provide evidence challenging the notion that *S. aureus* is a non-motile bacterium [81].

Spreading motility was found to be most like the sliding motility of bacteria. This mechanism is expressed by a colony of cells growing in number and producing surfactant to grow radially. Since this is a passive process, the surfactant blocks hydrogen bonding, thus reducing surface tension, which carries the bacteria cells outward. Combined with the passive movement of the growing colony of the cells in the middle of the aggregate pushing the outer cells further from the colony’s center, the bacteria can achieve motility [81].

Comet motility is more associated with the gliding motility of bacteria. This mechanism requires phenol-soluble modulins (PSM) surfactant, a known virulence factor [82,83]. *S. aureus* uses these PSM’s by producing them in their slime at the front of a spreading aggregate. This aggregate moves away from the main colony and leaves a slime trail behind it like a slug would. These aggregate offshoots are called pointed dendrites or microcolonies moving away from the colony’s center [82,83].

#### 4.5.1. Sortase A (SrtA)

Sortase A (SrtA) is an enzyme that plays a crucial role in *S. aureus* by anchoring surface proteins to the bacterial cell wall, including fibronectin-binding proteins (FnBPs) [84]. These surface proteins are essential for adhesion, biofilm formation, and motility, contributing to the bacterium’s ability to colonize host tissues and form protective biofilms. By anchoring these proteins, Sortase A facilitates the stable attachment of *S. aureus* to surfaces, making it a key player in infection establishment and persistence. Given its surface exposure, Sortase A is a promising antibody-based therapy target. Antibodies directed against Sortase A could inhibit its enzymatic activity, preventing the anchoring of virulence factors like FnBPs. This disruption could impair bacterial adhesion, motility, and biofilm formation, ultimately reducing the bacterium’s ability to cause infection [84]. Targeting Sortase A could, therefore, be a viable strategy to weaken *S. aureus*’s pathogenic capabilities.

#### 4.5.2. Fibronectin-Binding Proteins (FnBPs)

Fibronectin-binding proteins (FnBPs) play a crucial role in bacterial adhesion to host tissues and surfaces, which is a key step in colonization and infection. FnBPs mediate binding to fibronectin, a glycoprotein found in the extracellular matrix of host tissues, allowing *S. aureus* to firmly attach to host cells and medical devices [85]. The FnBPs facilitate the internalization of *S. aureus* by various human host cells, including osteoblasts, endothelial cells, and epithelial cells [86]. These proteins are also vital for biofilm formation, as they help bacteria adhere to surfaces and each other, establishing a protective bacterial community that resists immune responses and antibiotics.

Fibronectin Binding Protein A (FnBPA) and Fibronectin Binding Protein B (FnBPB) interact with fibronectin, while FnBPA also binds to fibrinogen [33]. The binding region consists of a highly conserved sequence of four repeat domains, each approximately 40 amino acids long. Fibronectin serves as a bridging molecule between *S. aureus* FnBPs and the host cell’s alpha-five beta-one integrins and heat-shock protein 60, which initiates signal transduction, activates tyrosine kinase and induces cytoskeletal rearrangements [87,88,89,90]. Large studies involving infected patients with *S. aureus* revealed that most strains carry both fnb genes; however, strains isolated from orthopedic implant-associated infections exhibited greater adherence to fibronectin compared to those from nasal carriers, endocarditis, septic arthritis, or osteomyelitis [88,89]. Mutant strains lacking either fnbA or fnbB did not show a decreased affinity for fibronectin, while only double-knockout mutants displayed significantly reduced fibronectin-binding activity [88].

Because FnBPs are surface-exposed, they present an excellent target for antibody-based therapies. Antibodies directed against FnBPs could block bacterial adhesion, prevent biofilm formation, and potentially reduce bacterial motility. Disrupting these interactions would impair *S. aureus’s* ability to colonize host tissues and spread, making FnBPs a promising therapeutic target for combating persistent infections, especially in cases involving implanted medical devices [91].

*S. aureus* demonstrates unexpected motility through spreading and comet formation, aiding in its survival and colonization. These forms of motility, facilitated by proteins SrtA and FnBPs, are integral to adhesion and biofilm formation, critical factors in the bacterium’s virulence. Targeting these surface-exposed proteins with antibodies could significantly impair motility, biofilm development, and overall infectivity. As research progresses, the list of surface-exposed proteins that serve as potential therapeutic targets will likely expand, offering new opportunities to combat *S. aureus* infections more effectively [85,91,92].

### 4.6. Resource Scavenging Molecules

*S. aureus* relies on sophisticated resource-scavenging systems to thrive in nutrient-limited environments, such as the human host. Key among these systems are their strategies for acquiring essential nutrients like iron and phosphate, which are critical for bacterial survival and virulence [93]. These resource acquisition mechanisms help *S. aureus* compete with host defenses and other microbes and represent potential targets for antibody-based therapies aimed at limiting the bacterium’s survival during infection.

#### 4.6.1. Iron Acquisition Program (Isd)

Bacterial survival within the human host depends on successfully acquiring nutrients, particularly iron. *S. aureus* secretes high-affinity iron-binding compounds (aureochelin and staphyloferrin) during iron starvation [94]. Upon sensing low iron, *S. aureus* initiates transcription of an iron acquisition program (Isd) that allows capture of heme and haptoglobin on the cell surface, transport of the iron-complex across the plasma membrane, and subsequent oxidative degradation of the heme within the cytoplasm [95]. Protein isdB is an iron-scavenging molecule anchored to the cell-wall that was identified as a potential vaccine candidate by Merck [96]. In the Phase IIb/III trial of vaccine candidate V710 (unadjuvanted isdB), termination of the study was recommended by the Data and Safety Monitoring Board due to safety concerns and low efficacy compared to the placebo in preventing *S. aureus* bacteremia or mediastinitis within 90 days post-operation. The safety concerns included an unexplained observance of higher multiorgan failure in the vaccine group [96].

#### 4.6.2. Glycerophosphoryl Diester Phosphodiesterase (GlpQ)

Due to its colonization of the nasal cavity in a significant portion of the population, *S. aureus* poses a considerable risk for invasive diseases. In this environment, *S. aureus* must compete with other bacteria, particularly for limited nutrients such as phosphate. Unlike commensal coagulase-negative staphylococci, *S. aureus* utilizes glycerophosphodiesters released by the secreted enzyme GlpQ from host lipids [97]. This enzyme allows *S. aureus* to cleave a variety of glycerol-3-phosphate (GroP) headgroups from deacylated phospholipids, making GlpQ a crucial component of the bacterium’s nutrient acquisition and a significant antigen during infection. GlpQ functions as a teichoicase, critical for accessing phosphate in low-nutrient environments. By mobilizing GroP from wall teichoic acids (WTAs) in commensal coagulase-negative staphylococci, *S. aureus* can thrive under phosphate-limiting conditions. This enzyme presents a promising target for polyclonal antibodies. Inhibiting GlpQ could restrict *S. aureus’s* phosphate sequestration capabilities, limiting its survival in competitive environments like the nasal cavity [97,98]

*S. aureus* employs sophisticated resource-scavenging systems, such as the Isd iron acquisition program and the GlpQ-mediated phosphate scavenging, to thrive in nutrient-poor environments like the human host. These mechanisms help the bacterium secure essential nutrients like iron and phosphate and offer potential targets for antibody-based therapies aimed at curbing bacterial survival during infection. As we understand these systems better, targeting key proteins involved in resource acquisition may present new strategies for limiting *S. aureus*’s virulence and persistence in the host.

### 4.7. Immunomodulators

*S. aureus* is highly adept at evading the host immune system, utilizing a range of immunomodulatory proteins to subvert immune responses, enhancing its survival, colonization, and virulence. These proteins interfere with the host’s ability to mount an effective defense, allowing the bacterium to persist in tissues and establish infections. Immunomodulators produced by *S. aureus* include proteins that block immune signaling pathways and enzymes that degrade key components of the immune response. By disrupting processes like neutrophil chemotaxis, antibody-mediated opsonization, and phagocytosis, these molecules significantly impair the host’s ability to clear infections. The strategic deployment of these proteins allows *S. aureus* to cause a range of infections, from minor skin issues to severe systemic diseases. As a result, these immunomodulatory proteins represent important targets for developing novel therapies to neutralize the bacterium’s virulence.

#### 4.7.1. Panton-Valentine Leukocidin (PV-L)

Panton-Valentine Leukocidin (PV-L) is a two-component toxin primarily associated with community-acquired methicillin-resistant *S. aureus* (CA-MRSA) strains [95]. It consists of two protein subunits, LukS-PV and LukF-PV, which work together to form pores in the membranes of host cells. It mainly targets polymorphonuclear leukocytes (PMNs) such as neutrophils. The binding of these subunits to the cell membrane leads to pore formation, cell lysis, and the release of pro-inflammatory mediators, contributing to tissue damage and the spread of infection. PVL is implicated in severe conditions, including necrotizing pneumonia, furunculosis, and osteomyelitis [99,100]. Regarding immunogenicity, PVL can induce an immune response; however, its widespread presence in virulent strains suggests that *S. aureus* can effectively use it to overcome immune challenges. Although PVL is secreted and extracellularly accessible, it has proven challenging as a drug target because blocking it alone does not always lead to successful infection control. Nonetheless, its role in severe infections makes it a potential multi-target therapeutic approach [95].

#### 4.7.2. Protein A

Protein A is a cell wall-associated protein in *S. aureus*. Protein A is encoded by the staphylococcal protein A (*spa*) gene and found in nearly all strains of the bacterium. This 40–60 kDa protein serves as a key immune evasion tool by binding to the Fc region of immunoglobulins (IgG1, IgG2) with high affinity and to IgM, IgA, and IgE with medium affinity [30]. By binding the Fcγ region of antibodies, Protein A disrupts proper opsonization and phagocytosis, preventing immune cells from recognizing and clearing the bacteria. Additionally, Protein A binds to VH3 B-cell receptors, functioning as a B-cell superantigen. This interaction triggers apoptosis in marginal zone B cells and B-1 cells, leading to reduced antibody production, further aiding *S. aureus* in immune evasion [101].

Protein A’s dual function in antibody binding and B-cell depletion makes it a potent virulence factor. Its immunogenicity prompted efforts to design vaccines targeting mutated forms of Protein A, which reduce its binding to antibodies and B-cell receptors, potentially enhancing immune recognition and clearance of *S. aureus*. These approaches aim to neutralize Protein A’s effects and reduce the bacterium’s ability to cause persistent infections [102].

#### 4.7.3. Nuc

Thermostable nuclease-like Nuc breaks down nucleic acids (DNA and RNA) and hydrolyzes them into smaller fragments [103]. This can disrupt cellular processes and contribute to tissue damage in infected hosts. Thermostable nuclease contributes to the survival of *S. aureus* by degrading extracellular DNA, which often traps bacteria in host tissues. This degradation enables the bacteria to evade immune detection and clearance. By targeting neutrophil extracellular traps (NETs), which are primarily composed of DNA, thermostable nuclease further disrupts the host’s defense mechanisms. Additionally, it enhances bacterial virulence by breaking down nucleic acids and facilitating the spread of the bacteria within the host. It is also a marker for identifying *S. aureus* in laboratory settings. Diagnostic use: due to the enzyme’s unique thermostable properties, the thermostable nuclease test (TNase test) is widely used to confirm *S. aureus* in clinical samples [103].

#### 4.7.4. Chemotaxis Inhibitory Protein (CHIP)

Chemotaxis Inhibitory Protein (CHIP) is an extracellular protein that blocks neutrophil chemotaxis and immune cell recruitment. CHIP works by inhibiting the Formyl Peptide Receptor (FPR) and the C5a receptor (C5aR), both critical for neutrophil migration towards infection sites [104]. By binding to these receptors, CHIP prevents activation by the body’s natural ligands, halting neutrophil chemotaxis and interfering with the host’s inflammatory response. CHIP’s ability to inhibit C5aR, a receptor activated by the complement system, and FPR, which responds to bacterial peptides, allows *S. aureus* to evade the immune system by limiting the host’s early immune response [105]. Due to its extracellular location, CHIP is accessible to antibodies and represents a promising therapeutic target, but its moderate immunogenicity may limit immediate immune recognition. CHIP has moderate immunogenicity, likely because CHIP works quickly to block immune signaling before a robust immune response can develop, reducing the likelihood of prolonged exposure and immune activation [106].

#### 4.7.5. Extracellular Adherence Protein (EAP)

Extracellular Adherence Protein (EAP) plays a pivotal role in the virulence of *S. aureus*. It is implicated in multiple stages of infection, primarily by facilitating the internalization of *S. aureus* into eukaryotic cells and modulating the host immune response [107,108]. EAP exhibits extensive binding capabilities, allowing it to interact with host cell surfaces and impair neutrophil recruitment and diapedesis. Specifically, EAP disrupts neutrophil migration by binding to lymphocyte function-associated antigen-1 (LFA-1) on neutrophils, which blocks their interaction with intercellular adhesion molecule-1 (ICAM-1) on endothelial cells. This interaction inhibits neutrophil extravasation, weakening the host’s immune response at the infection site. By blocking these critical immune pathways, EAP enables *S. aureus* to evade immune detection and clearance, contributing to its persistence and enhanced virulence in host tissues [108,109].

*S. aureus* employs a wide array of sophisticated immunomodulatory proteins to evade host immune responses, enhancing its ability to colonize and cause disease. These proteins, including Panton-Valentine Leukocidin (PVL), Protein A, thermostable nuclease (Nuc), Chemotaxis Inhibitory Protein (CHIP), and Extracellular Adherence Protein (EAP), target critical aspects of the immune system such as neutrophil function, antibody-mediated responses, and cellular signaling pathways. By disrupting key immune processes—such as neutrophil chemotaxis, phagocytosis, and immune cell recruitment—these virulence factors enable *S. aureus* to establish persistent infections and resist clearance by the host. Understanding these mechanisms opens new therapeutic avenues, such as developing polyclonal antibodies or inhibitors that target these immunomodulators. These treatments could complement existing antibiotics and potentially prevent biofilm formation or enhance the immune system’s ability to combat this formidable pathogen.

### 4.8. Other Membrane Biomolecules

*S. aureus* employs various membrane-associated biomolecules essential to its virulence, survival, and ability to form biofilms. These biomolecules contribute to key processes such as adhesion to host tissues, immune evasion, and biofilm formation, which are crucial in chronic infections and antibiotic resistance. The cell surface proteins and polysaccharides of *S. aureus* facilitate colonization and protect the bacteria from immune detection and antimicrobial treatments. This section explores the diverse roles of biofilm-associated proteins (Bap), capsule polysaccharides, and poly-N-acetylglucosamine (PNAG), among other membrane components, in the pathogenesis of *S. aureus* infections. Understanding these molecules provides valuable insights into potential therapeutic targets for preventing biofilm formation and enhancing immune responses, thereby offering new avenues for combating multidrug-resistant *S. aureus* infections.

#### 4.8.1. Biofilm Associated Protein (Bap)

Bap promotes biofilm development in *S. aureus* strains. Bap is a 239 kDa covalently linked cell surface protein with structural components similar to MSCRAMMs [33]. Bap plays an important role in forming and maintaining biofilms, providing a protective environment for the bacteria. Biofilms allow *S. aureus* to adhere to surfaces, resist immune responses, and survive in hostile environments, contributing to persistent infections and antibiotic resistance [63,110]. Developing polyclonal antibodies against Bap could offer a method to prevent biofilm formation or disrupt early biofilm development.

#### 4.8.2. Capsule Polysaccharide (Types 1, 2, 5, and 8)

*S. aureus* produces various polysaccharide capsules that significantly affect immune evasion. Compared to other pathogens like *Haemophilus influenzae type b* and *Streptococcus pneumoniae*, *S. aureus* sheds its capsule less frequently, with types 1 and 2 being heavily encapsulated and associated with a mucoid colony morphology, although these rarely cause clinical disease [111,112].

However, polysaccharide types 5 and 8 are more clinically relevant, accounting for about 85% of isolates found in clinical samples [113]. Antibodies specific to these two polysaccharides have shown potential in promoting type-specific opsonophagocytosis and conferring some protection in animal models. However, further studies are needed to confirm their effectiveness in humans. Vaccines targeting types 5 and 8 have been tested in clinical trials, where the results demonstrated that the vaccine was well tolerated, but it failed to produce significant efficacy in the long term [113]. Modest efficacy was noted between the 30- to 40-week timepoints, but no efficacy was observed after 50 weeks.

The limited success of these vaccines could be attributed to various factors, such as the USA300 strain not shedding either type 5 or 8 capsules or the possibility that these capsules do not function effectively as protective antigens [114]. Additionally, the immune-compromised status of participants in the trials may have contributed to insufficient antibody production.

Two multi-antigen vaccines, SA3Ag and SA4Ag, showed stronger immunogenicity than StaphVAX. SA4Ag, which includes recombinant P305A, induced robust immune responses but failed to prevent postoperative *S. aureus* infections in a phase III trial (0.0% efficacy) [4,115]. Infection and colonization rates were similar between groups, despite strong antibody responses. SA4Ag was safe and well tolerated but not effective in preventing infection.

One possible path forward is the incorporation of capsular polysaccharides in multi-component vaccines, similar to current *S. aureus* vaccines in development by Pfizer and GlaxoSmithKline [116]. These vaccines combine capsular polysaccharides with other antigens like clumping factor A and alpha-toxoid, which may enhance immune response and provide broader protection. Interestingly, the expression of type 5 or 8 capsules has been found to inversely correlate with the production of poly-N-acetylglucosamine (PNAG), suggesting that in vivo, *S. aureus* may preferentially express PNAG, further complicating the development of capsule-based vaccines [30].

#### 4.8.3. Poly-N-acetylglucosamine (PNAG)

PNAG is an intercellular adhesin first identified in coagulase-negative *Staphylococcus aureus* (CoNS) strains [117]. This polysaccharide plays a key role in biofilm formation by mediating cell-to-cell adhesion, a critical process in the persistence of bacterial infections. The synthesis of PNAG is controlled by the *icaADBC* gene locus, which is common in CoNS strains but also present in pathogenic *S. aureus*. PNAG has been explored as a potential target for immune-based therapies in experimental studies. In one study, rabbits were immunized with purified PNAG, producing anti-PNAG IgG antibodies [117]. When these antibodies were passively transferred to mice, they significantly reduced bacterial burden following a kidney infection model caused by *S. aureus*. This suggests that targeting PNAG could help control *S. aureus* infections by preventing biofilm formation and enhancing bacterial clearance. Further research has explored the use of human monoclonal antibodies directed against deacetylated PNAG epitopes (dPNAG) [118]. These antibodies were shown to be opsonophagocytic, meaning they helped the immune system recognize and destroy bacteria. In mouse models, anti-dPNAG antibodies provided modest protection, indicating that PNAG could be a viable target for immunotherapy. A 2016 clinical trial was launched to explore these findings further [118]. In conclusion, a therapeutic approach based on the passive administration of anti-PNAG IgG antibodies or human monoclonal antibodies directed towards dPNAG is an idea that requires further exploration in human clinical trials.

#### 4.8.4. Lipoteichoic Acid (LTA)

Lipoteichoic acid (LTA) is a charged polymer in the *S. aureus* cell wall, composed of D-alanine and N-acetylglucosamine, and anchored to the cytoplasmic membrane via a glycolipid [33,119]. LTA plays a crucial role in adhesion and biofilm formation. The enzyme DltA, which transfers D-alanine to LTA, helps modulate its charge; mutations in dltA result in a more negatively charged LTA and impaired biofilm formation. Similarly, mutations in the ypfP gene reduce LTA production and biofilm formation. Targeting LTA biosynthesis, such as inhibiting DltA or YpfP, may offer strategies to reduce biofilm-associated infections. However, attempts to use LTA as a vaccine target, such as with the anti-LTA monoclonal antibody pagibaximab, have not proven effective in clinical trials, particularly in neonates [120].

#### 4.8.5. Wall Teichoic Acid (WTA)

Wall Teichoic Acid (WTA) is a polymer consisting of alternating phosphate and ribitol groups covalently linked to the peptidoglycan layer in the *S. aureus* cell wall [121]. WTA plays a crucial role in bacterial cell shape, regulation of autolytic enzymes, and adherence to host tissues, contributing to the organism’s ability to form biofilms and evade immune defenses [119]. Targeting WTA biosynthesis is being explored as a potential strategy for combating *S. aureus* infections, as disrupting its production can weaken the bacterial cell wall and reduce virulence.

The membrane biomolecules of *S. aureus*, including the Biofilm Associated Protein (Bap), Capsule Polysaccharides, Poly-N-acetylglucosamine (PNAG), Lipoteichoic Acid (LTA), and Wall Teichoic Acid (WTA), play pivotal roles in biofilm formation, immune evasion, and colonization. These molecules are integral to the pathogen’s survival and pathogenicity, contributing to the formation of protective biofilms, inhibition of immune recognition, and resistance to therapeutic interventions. Investigations into biofilm dispersal using proteinase K and the development of polyclonal antibodies targeting these biomolecules have demonstrated promising therapeutic potential. Moreover, while still in early stages of development, polysaccharide-based vaccines offer hope for long-term protection against *S. aureus* infections. Strategies aimed at inhibiting these membrane-associated components, such as LTA and WTA, or enhancing immune responses through multi-component vaccines targeting key molecules like PNAG, represent the future of combating *S. aureus* infections, particularly in the context of multidrug resistance and chronic biofilm-associated diseases. By disrupting these crucial molecular processes, new therapies may provide a more effective, long-lasting defense against this formidable pathogen. [122,123].

## 5. Discussion

*S. aureus* can rapidly develop antibiotic resistance through various mechanisms, including gene mutations, horizontal gene transfer, and biofilm formation. One of the most notorious examples is methicillin-resistant *S. aureus* (MRSA), which evolved resistance to beta-lactam antibiotics shortly after their introduction. Studies have shown that *S. aureus* can acquire resistance within a few years of antibiotic use due to its ability to modify target sites, produce enzymes like beta-lactamase, and form biofilms that protect it from drugs [124,125]. This adaptability allows it to withstand new antibiotics quickly, making it a significant public health concern.

Developing antibodies targeting specific bacterial components offers a promising approach to combating resistant infections [4]. Therapeutic antibodies engage the immune system’s effector mechanisms, including neutralizing toxins, enhancing phagocytosis, activating complement pathways, and promoting T and B cell responses. These mechanisms can provide a multifaceted defense against pathogens, offering enhanced specificity and reducing the potential for bacterial escape. Combination therapies with numerous monoclonal antibodies, which target multiple targets, offer broader coverage and reduce the likelihood of resistance developing compared to monoclonal antibodies [27].

A critical aspect to consider moving forward is the method of drug delivery. As discussed in this review, the potential applications for polyclonal therapies are varied, spanning various environments and infection sites. In many instances, current antibiotic concentrations in serum or infection sites are below the minimum inhibitory concentration (MIC), diminishing treatment efficacy and raising the risk of resistance. Ideally, the delivery route should be tailored to the infection site. For bacteremia, intravenous administration remains the best option. However, a more targeted approach, such as delivering IgA/IgG via nebulizer, may be more effective for conditions such as pneumonia or cystic fibrosis. Studies have shown that nebulized immunoglobulins can boost levels in bronchoalveolar lavage fluid and confer protection for up to 48 h. Additionally, topical applications could be helpful for infections like keratitis or folliculitis. Recent findings also highlight that antibodies can enhance drug efficacy by modifying bacterial cell surfaces when combined with antibiotics, resulting in synergistic antimicrobial effects [126,127].

*S. aureus* is a notorious pathogen responsible for a range of diseases, from skin infections to severe conditions like necrotizing pneumonia, endocarditis, and sepsis, and continues to evolve and develop resistance to antibiotics. The human immune system combats these infections on multiple fronts, but *S. aureus* employs various immune evasion tactics, including producing immunomodulatory proteins such as Protein A, which disrupts antibody-mediated responses, and CHIP, which blocks immune cell recruitment. To effectively combat these adaptations, combination antibody therapies targeting multiple virulence factors offer a promising solution. These therapies could be tailored to disrupt critical bacterial processes such as immune evasion, biofilm formation, and resource scavenging, improving treatment outcomes and reducing the development of resistance.

As the genomic landscape of *S. aureus* is further studied and understood, these antibody combinations can be continuously refined and personalized. Given the rapid evolution of *S. aureus*, antibody technologies must evolve in parallel, leveraging advanced engineering techniques to stay ahead of bacterial adaptations. This approach addresses the challenges posed by multidrug-resistant strains and helps mitigate future risks, potentially reducing the global burden of antibiotic-resistant *S. aureus* infections.

## Figures and Tables

**Figure 1 antibiotics-13-01046-f001:**
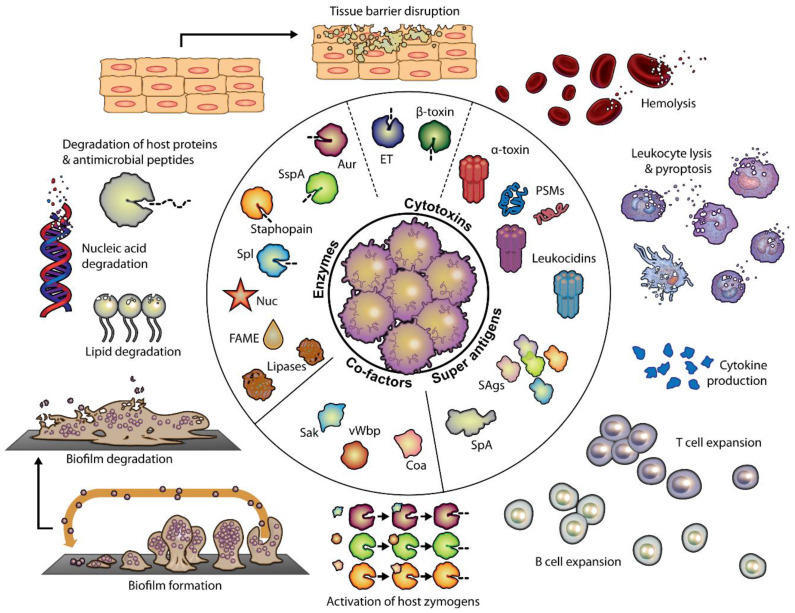
Virulence factors of *Staphylococcus aureus* and their role in host–pathogen interactions. This figure illustrates the diverse virulence factors produced by *S. aureus*, emphasizing their contribution to tissue invasion, immune evasion, and biofilm formation. Cytotoxins, such as α-toxin, β-toxin, and phenol-soluble modulins (PSMs), disrupt host cell membranes, leading to hemolysis and leukocyte lysis via pore formation. Leukocidins and superantigens (SAgs) further exacerbate host immune responses by inducing leukocyte pyroptosis and cytokine storms, driving inflammation. Enzymes like staphopain, lipases, and nucleases (Nuc) degrade host proteins, lipids, and nucleic acids, promoting tissue invasion. Other virulence factors, such as coagulases (Coa) and staphylococcal protein A (SpA), aid in immune evasion by blocking phagocytosis or activating host zymogens. Biofilm formation is depicted at the bottom left, illustrating how *S. aureus* biofilms contribute to persistent infections by shielding bacterial colonies from host defenses and antibiotics.

**Figure 2 antibiotics-13-01046-f002:**
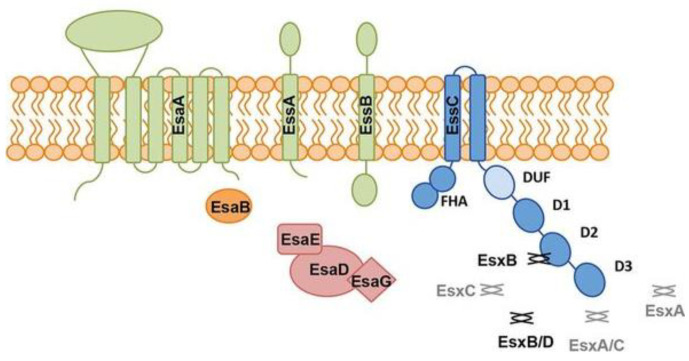
Model of the T7SS machinery in *S. aureus*, spanning the inner membrane. EssA, EssB, and EssC form the core membrane complex. These integral membrane proteins create the secretion channel. EssC, an ATPase, drives the export of effector proteins, including EsxB, EsxC, and EsxD, which are secreted into the extracellular space. The EsaE chaperone facilitates the secretion of EsaD, a nuclease toxin, which is neutralized within the bacterial cell by EsaG to prevent self-damage. EsaB acts as a negative regulator, preventing secretion when not required. The extracellular secretion of effectors enables *S. aureus* to engage in bacterial competition and evade host immune responses.

**Table 1 antibiotics-13-01046-t001:** Summary of the key drugs used to treat MRSA infections from IDSA guidelines [24].

Drug	Type	Use	Mechanism
Vancomycin	Glycopeptide	First-line for severe MRSA	Inhibits cell wall synthesis
Daptomycin	Lipopeptide	Bacteremia, endocarditis	Disrupts bacterial membrane
Teicoplanin	Glycopeptide	Alternative to vancomycin (not in the US)	Inhibits cell wall synthesis
Ceftaroline	Cephalosporin	Alternative for bacteremia	Binds altered penicillin-binding proteins
Linezolid	Oxazolidinone	Pneumonia, soft tissue infections	Inhibits protein synthesis
Dalbavancin	Glycopeptide (Long-acting)	Outpatient care for skin infections	Inhibits cell wall synthesis
Oritavancin	Glycopeptide (Long-acting)	Outpatient skin infections	Disrupts cell wall and membrane

## Data Availability

No new data were created or analyzed in this study.

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
