# Peer review of "Potential Therapeutic Targets for Combination Antibody Therapy Against *Staphylococcus aureus* Infections"

_antibiotics, 2024, doi:10.3390/antibiotics13111046_

Round 1

Reviewer 1 Report

Comments and Suggestions for Authors

Comments and Suggestions for the Authors

  The ‘Review’ manuscript “Potential Therapeutic Targets for Combination Antibody Therapy against Staphylococcus aureus Infections” is well written and a comprehensive overview of Staphylococcus aureus virulence factors and their potential as therapeutic targets for combination antibody therapy. This review is detailed, well-structured, and delivers relevant information to support potential therapeutic approach. The manuscript is adequately detailed to address the goal set by the authors. I recommend the manuscript to be Accepted with few Minor modifications/rectifications before publication. I have a few minor comments and suggestions to further enhance the manuscript.

·        Title reflects the main subject of the manuscript.

·    Abstract is good, that summarizes and reflects the topic described in the manuscript.

·        Key words are adequate.

·        The introduction is adequate and comprehensive.

·        Line 115; Reference for Norvancomycin should be the following

Li, J., He, S., Yang, Z., and Lu, C. (2017). Pharmacokinetics and cerebrospinal fluid penetration of norvancomycin in Chinese adult patients. Int. J. Antimicrob. Agents 49, 603–608. doi: 10.1016/j.ijantimicag.2017.01.014

Reference 16 (Guo et. al., 2020; doi:10.3389/fcimb.2020.00107) themselves cited the above-mentioned article as they talk about Norvancomycin.

·     Figure 1 is a beautiful illustration of existing knowledge about Staphylococcal virulence factors.

·       Line 246- 251, Line 252-260; As the whole paragraphs were based on single reference, mentioning Reference 26 only at the end of paragraphs is enough.  

·        Line 266 -271; Mention Reference.

·   Authors have effectively proposed the targeting of virulence factors in Staphylococcus aureus for combination antibody therapy. To strengthen their argument (what I feel), authors could consider including examples of existing vaccines or antibody therapies that successfully target virulence factors in other bacterial pathogens. This approach would provide context and evidence for the feasibility and effectiveness of such proposals made by the authors.

·        References are adequate.

Author Response

Comments 1: Line 115; Reference for Norvancomycin should be the following: 

Li, J., He, S., Yang, Z., and Lu, C. (2017). Pharmacokinetics and cerebrospinal fluid penetration of norvancomycin in Chinese adult patients. Int. J. Antimicrob. Agents 49, 603–608. doi: 10.1016/j.ijantimicag.2017.01.014

Reference 16 (Guo et. al., 2020; doi:10.3389/fcimb.2020.00107) themselves cited the above-mentioned article as they talk about Norvancomycin.

Response 1: Thank you for pointing this out. We have changed the reference for line 115 from 16 Guo et al. to 19 Li et al. 

Comments 2: Line 246- 251, Line 252-260; As the whole paragraphs were based on single reference, mentioning Reference 26 only at the end of paragraphs is enough.  

Response 2: Thank you for pointing this out. We have made the following changes: Line 246-251, Line 252-260 were modified to only mention Reference 27 at the end of the paragraphs.

Comments 3: Line 266 -271; Mention Reference.

Response 3: Thank you for pointing this out. We have made the following changes: Line 266-271, Reference 27 Dinges et al. has been added.

Comments 4: To strengthen their argument (what I feel), authors could consider including examples of existing vaccines or antibody therapies that successfully target virulence factors in other bacterial pathogens. This approach would provide context and evidence for the feasibility and effectiveness of such proposals made by the authors.

Response 4: Thank you for the recommendation. We agree and have taken the following steps: a table of S. aureus vaccines with clinical trial numbers has been added to the discussion section (Table 2) and descriptions of other approaches attempted to create an effective vaccine. 

Reviewer 2 Report

Comments and Suggestions for Authors

In their manuscript, Ke et al. present a comprehensive and exhaustive review of current treatments for Staph infections. I learned a lot and am willing to recommend publication eventually. However, I still have several suggestions for improving the manuscript:

In the abstract, the authors promise that they will discuss "a promising approach that combines monoclonal antibodies targeting multiple S. aureus epitopes, offering synergistic efficacy in treating infections." Unfortunately, there was very little about this -- basically one page out of a 20-page narrative. I recommend that the authors retitle the "Discussion" section and label it "Synergistic Monoclonal Antibody Therapy" or something like that and make very specific references to that approach. What is the history of the approach? The authors discuss numerous molecular targets in their review. Are any of these targets for synergistic monoclonal antibody therapy? Which ones? If none have been tested, which ones should be tested? (I could not tell if this was simply a conceptual strategy or whether or not there have been successes already.) Especially given the title, I would like the authors to propose criteria for choosing molecular targets for monoclonal antibody therapy.

Can they come up with a visualization that illustrates the novelty of this approach?

It is a very promising manuscript, but the last page of the story should be expanded and developed before publication.

Finally, some more minor details:

What are the PRRs for Staph?  (Lines 67-74)

What are the known mechanisms of drug resistance for MRSA? (Lines 98-147)

Citations should be added to the following paragraphs (they only have a citation at the end of the paragraph). If the citations at the end of the paragraph refer to earlier statements as well, they should also be included there:

Lines 200-208

Lines 217-225

Lines 226-233

Lines 365-372

Lines 373-380

Lines 398-405

Lines 778-790

Lines 884-897

Author Response

Comments 1: I recommend that the authors retitle the "Discussion" section and label it "Synergistic Monoclonal Antibody Therapy" or something like that and make very specific references to that approach.

Response 1: Thank you for the recommendation. We changed the Discussion section title to the one suggested. 

Comments 2: What is the history of the approach? The authors discuss numerous molecular targets in their review. Are any of these targets for synergistic monoclonal antibody therapy? Which ones? If none have been tested, which ones should be tested? (I could not tell if this was simply a conceptual strategy or whether or not there have been successes already.) 

Response 2: Thank you for your comments. We took the following steps to clarify the purpose of discussing numerous molecular targets and our intent in suggesting synergistic monoclonal antibody therapy: in the discussion section, a table of previous staph aureus vaccines that undergone clinical trials has been added along with descriptions of previous engineering methods trialed to create an effective vaccine. 

Comments 3: 

What are the PRRs for Staph?  (Lines 67-74)

Response 3: Thank you for the clarification question. We decided to leave it as is since we feel that the current sentence "macrophages and DCs are activated by pattern recognition receptors (PRRs), including toll-like receptors that recognize cell wall biomolecules and organelles glycoproteins, carbohydrates, and flagella" already include an example of PRRs. 

Comments 4: What are the known mechanisms of drug resistance for MRSA? (Lines 98-147)

Response 4: Thank you for the question. We agree and added a sentence giving a brief overview of possible resistance mechanisms (lines: 103-106 is in red). 

Comments 5: Citations should be added to the following paragraphs (they only have a citation at the end of the paragraph). If the citations at the end. of the paragraph refer to earlier statements as well, they should also be included there: 

Lines 200-208

Lines 217-225

Lines 226-233

Lines 365-372

Lines 373-380

Lines 398-405

Lines 778-790

Lines 884-897

Response 5: Thank you for the recommendation. The following changes have been made and for clarification, the lines above have been shifted due to adding more lines earlier in the manuscript. So any added references have been marked in red.

Lines 200-208 -> 205-213. Reference 32 has been added to the beginning sentence. 

Lines 217-225 -> 222-230. Reference 28 has been added to the beginning sentence. 

Lines 226-233 -> 231-238. Reference 28, 30 has been added to the beginning sentence. 

Lines 365-372 -> 370-377. Reference 56 has been added to the beginning sentence. 

Lines 373-380 -> 378-385. Reference 56 has been added to the beginning sentence. 

Lines 398-405 -> 403-410. No changes were made because it is our contribution. 

Lines 778-790 -> 784-795. No changes were made because it is our summary for the section. 

Lines 884-897 -> 889-904. References 1244 and 125 have been added to the beginning sentence. 

Thank you for taking the time to read and edit this manuscript. We appreciate your time and thoughts. 

Reviewer 3 Report

Comments and Suggestions for Authors

The manuscript by Sharon Ke et al. entitled ‘Potential therapeutic targets for combination antibody therapy against Staphylococcus aureus infections’ reviewed the clinical treatment methods against S. aureus infections, and proposed the potential targets for the development of potent antibiotics and antibodies, and discussed the combination antibody therapy. The manuscript provides a comprehensive introduction to the S. aureus infections and describes the prospective targets on which we should focus, however, before it can be published, there are several comments and questions that need to be addressed or corrected.

1.        The title of this manuscript is not entirely accurate. The manuscript begins with a brief introduction to S. aureus and infections and followed by a summary of the treatment strategy, and a list of prospective targets. The combination of antibody therapy is only mentioned in the discussion section. Thus, if the authors want to talk about the combination antibody therapy, the combination antibody therapy should be included in the main section, at least the current status or circumstance should be discussed in the main section.

2.        The classification in Figure 1 is very confusing and difficult to align with the 8 classes in Section 4. Additionally, what’s the difference between the dashed and the solid lines inside the circle of Figure 1?

3.        In Figure 2, it would be clear if the ATP to ADT transformation were added to Figure 2 to indicate that the progress requires energy.                  

4.        Use vivid and fascinating descriptions in the writing. The manuscript's description is a tad dry, especially Section 4, which lacks figures or colorful descriptions and is difficult to understand. I suppose you can select a few targets to cover in-depth, such as the Sec pathway, which I believe you have already done, or you can discuss how these targets are now blocked or used.

Author Response

Comments 1: 

The title of this manuscript is not entirely accurate. The manuscript begins with a brief introduction to S. aureus and infections and followed by a summary of the treatment strategy, and a list of prospective targets. The combination of antibody therapy is only mentioned in the discussion section. Thus, if the authors want to talk about the combination antibody therapy, the combination antibody therapy should be included in the main section, at least the current status or circumstance should be discussed in the main section.

Response 1: Thank you for your suggestions. We agree that the topic of combination antibody therapy can be expanded. The changes we made are to include previous staph aureus vaccines that have undergone clinical trials and their mechanisms. A table was added to illustrate this (Table 2). 

Comments 2: 

The classification in Figure 1 is very confusing and difficult to align with the 8 classes in Section 4. Additionally, what’s the difference between the dashed and the solid lines inside the circle of Figure 1?

Response 2: Thank you for your suggestions. There are no difference between the dashed and the solid lines inside the circle of Figure 1. We decided to leave the Figure as is since it is meant to be a general overview and not correlate directly with section 4. 

Comments 3: In Figure 2, it would be clear if the ATP to ADT transformation were added to Figure 2 to indicate that the progress requires energy.

Response 3: Thank you for your suggestions. We decided not to add ATP and ADT transformation to the figure to keep it clean. However, we did upgrade the graphic for the figure to be more eye-catching. 

Comments 4: Use vivid and fascinating descriptions in the writing. The manuscript's description is a tad dry, especially Section 4, which lacks figures or colorful descriptions and is difficult to understand. I suppose you can select a few targets to cover in-depth, such as the Sec pathway, which I believe you have already done, or you can discuss how these targets are now blocked or used.

Response 4: Thank you for your suggestions. We added how some of the targets have been utilized in past staph aureus vaccines in the last section. 

Thank you for taking the time to read over our manuscript and provide feedback. We appreciate your time and expertise. 

Round 2

Reviewer 3 Report

Comments and Suggestions for Authors

Thank you for addressing each of my questions individually. I agree that the work has met the journal's publishing standard. I have an additional suggestion for the manuscript's abstract for your reference. The abstract, as the most important portion of the document, should lead to the content in a short and precise description, as well as an accurate description of the full manuscript. The first two-thirds of the abstract now depict the status of S. aureus, I think it would be better to state the current status of S. aureus in the introduction, and instead, the biological properties of S. aureus can be put in the abstract.

Author Response

Comment 1: The abstract, as the most important portion of the document, should lead to the content in a short and precise description, as well as an accurate description of the full manuscript. The first two-thirds of the abstract now depict the status of S. aureus, I think it would be better to state the current status of S. aureus in the introduction, and instead, the biological properties of S. aureus can be put in the abstract.

Response 1: Thank you for the additional suggestions on the abstract. We have made the following addition to the abstract to tie in the biological properties of S. aureus: "Known for its array of virulence factors, including surface proteins that promote adhesion to host tissues, enzymes that break down host barriers, and toxins that contribute to immune evasion and tissue destruction, S. aureus poses a serious health threat".